# *Mycobacterium abscessus* Opsonization Allows an Escape from the Defensin Bactericidal Action in *Drosophila*

Hamadoun Touré,[a] Nicolas Durand,[a] Isabelle Guénal,[b] Jean-Louis Herrmann,[a,c] Fabienne Girard-Misguich,[a] Sébastien Szuplewski[b]

[a]Université Paris-Saclay, UVSQ, INSERM, Infection et Inflammation, Montigny-Le-Bretonneux, France
[b]Université Paris-Saclay, UVSQ, LGBC, Versailles, France
[c]Assistance Publique-Hôpitaux de Paris, Hôpitaux Universitaires Ile-de-France Ouest, GHU Paris-Saclay, Hôpital Raymond Poincaré, Garches, France

Fabienne Girard-Misguich and Sébastien Szuplewski contributed equally to this article. They are co-senior authors.

**ABSTRACT** *Mycobacterium abscessus*, an intracellular nontuberculous *mycobacterium*, is considered the most pathogenic species among the group of rapidly growing mycobacteria. The resistance of *M. abscessus* to the host innate response contributes to its pathogenicity in addition to several virulence factors. We have recently shown in *Drosophila* that antimicrobial peptides (AMPs), whose production is induced by *M. abscessus*, are unable to control mycobacterial infection. This could be due to their inability to kill mycobacteria and/or the hidden location of the pathogen in phagocytic cells. Here, we demonstrate that the rapid internalization of *M. abscessus* by *Drosophila* macrophages allows it to escape the AMP-mediated humoral response. By depleting phagocytes in AMP-deficient flies, we found that several AMPs were required for the control of extracellular *M. abscessus*. This was confirmed in the Tep4 opsonin-deficient flies, which we show can better control *M. abscessus* growth and have increased survival through overproduction of some AMPs, including Defensin. Furthermore, Defensin alone was sufficient to kill extracellular *M. abscessus* both *in vitro* and *in vivo* and control its infection. Collectively, our data support that Tep4-mediated opsonization of *M. abscessus* allows its escape and resistance toward the Defensin bactericidal action in *Drosophila*.

**IMPORTANCE** *Mycobacterium abscessus*, an opportunistic pathogen in cystic fibrosis patients, is the most pathogenic species among the fast-growing mycobacteria. How *M. abscessus* resists the host innate response before establishing an infection remains unclear. Using *Drosophila*, we have recently demonstrated that *M. abscessus* resists the host innate response by surviving the cytotoxic lysis of the infected phagocytes and the induced antimicrobial peptides (AMPs), including Defensin. In this work, we demonstrate that *M. abscessus* resists the latter response by being rapidly internalized by *Drosophila* phagocytes. Indeed, by combining *in vivo* and *in vitro* approaches, we show that Defensin is able to control extracellular *M. abscessus* infection through a direct bactericidal action. In conclusion, we report that *M. abscessus* escapes the host AMP-mediated humoral response by taking advantage of its internalization by the phagocytes.

**KEYWORDS** *Mycobacterium abscessus*, phagocytes, opsonization, Defensin, *Drosophila*

**M**ycobacteria form a complex group, with a large majority of saprophytic mycobacteria that are an integral part of our daily environment. A minority of strictly pathogenic mycobacteria for humans and animals, for whom each acquisition corresponds to an infectious process or any case, is evidenced by a host response to the infection (1, 2). Different sociodemographic and lifestyle conditions (e.g., showering, smoking), increasing age of the general population, increasing frequency of chronic respiratory diseases, and complex therapeutic and clinical management are the bedrock of long-lasting infections that are difficult to treat and linked to so-called opportunistic mycobacteria (2). Among these are the

Address correspondence to Fabienne Girard-Misguich, fabienne.misguich@uvsq.fr, or Sébastien Szuplewski, sebastien.szuplewski@uvsq.fr.

The authors declare no conflict of interest.

*Mycobacterium avium* complex and the *Mycobacterium abscessus* complex, the most frequently found mycobacteria in the context of bronchiectasis or cystic fibrosis (3, 4).

*M. abscessus* has emerged over the last few decades as a difficult pathogen to eradicate due to its intrinsic multidrug resistance and the deleterious nature of pulmonary infection caused by this bacterium (2, 5). The particularity of this *mycobacterium* is that it belongs to the predominant group of fast-growing mycobacteria, most of which are saprophytic, such as *Mycobacterium smegmatis*. Several recent works have shown that *M. abscessus* possesses properties similar to those described in pathogenic mycobacteria such as *Mycobacterium tuberculosis* and which would account for the increased infectivity and virulence of *M. abscessus* compared to the other rapidly growing mycobacteria (RGM) (6, 7).

We have recently described an additional property, also found in *M. tuberculosis*, i.e., the ability to resist and survive after lysis of infected macrophages by cytotoxic NK cells, thus allowing it to resist the innate response of the host and survive within it (8). This resistance to phagocyte lysis was demonstrated using a thoracic injection model of *M. abscessus* infection in *Drosophila melanogaster*, thus avoiding the use of excessively large animals, and was confirmed with primary *M. abscessus*-infected murine macrophages with autologous NK cells (8).

Indeed, *Drosophila* is sensitive to intrathoracic injection of the smooth morphotype of *M. abscessus* (S-*M. abscessus*) at low doses of about 10 to 100 bacteria, resulting in death of flies almost between 6 and 10 days postinfection (dpi). S-*M. abscessus* is able to multiply within *Drosophila* phagocytic plasmatocytes (the equivalent of macrophages), confirming its propensity to resist the consequences of phagocytosis as demonstrated in other cellular models (6, 7). The death of flies was also a result of the resistance of *M. abscessus* to lysis and caspase-dependent apoptotic cell death of *Drosophila*-infected phagocytic plasmatocytes, the predominant cell type within *Drosophila* immune blood cells (8).

Phagocytic plasmatocytes transiently control S-*M. abscessus* infection in *Drosophila* as demonstrated, on one hand, by increased mortality of the flies depleted from this population upon infection and, on the other hand, by increased resistance to infection of flies maintaining this population in the absence of thanacytes (8). Indeed, thanacytes are a subpopulation of *Drosophila* immune blood cells (hemocytes), which potentially play the role of NK cells found in humans (8, 9).

Knowing that phagocytic plasmatocytes are permissive to S-*M. abscessus* infection, the resistance of flies could not be linked only to this mechanism. It might integrate the humoral response of flies, as we previously demonstrated antimicrobial peptide (AMP)-encoding genes' induction during the infection by *M. abscessus* (8). AMPs are the main effectors of *Drosophila* humoral immune response, which is activated after recognition of pathogen-associated molecular patterns by pattern recognition receptors (10). This response is mediated by two NF-$\kappa$B pathways, the Toll pathway, mainly activated by fungi and Gram-positive bacteria, and the immune deficiency (Imd) pathway, mainly activated by Gram-negative bacteria (11). AMPs are small positively charged peptides that bind to the hydrophobic regions of microbial membranes, creating pores, destabilizing membrane integrity, and ultimately causing pathogen death (12).

The purpose of this work was to explore how S-*M. abscessus* resists *Drosophila* AMP-mediated humoral response. We demonstrate that extracellular S-*M. abscessus* can be controlled during infection by certain AMPs. To resist them, the bacterium must be present inside plasmatocytes. Indeed, its rapid opsonization and internalization by plasmatocytes offer S-*M. abscessus* protection from the AMPs, particularly the direct bactericidal activity of Defensin. Here, we highlight that S-*M. abscessus* gains an advantage from its internalization to protect itself and resist the host AMP humoral response.

## RESULTS

***In vivo*** **evidence of the role of Defensin and Toll-regulated AMPs in the control of** ***M. abscessus*** **infection in immunocompromised flies.** The resistance of flies during the course of S-*M. abscessus* infection or in the absence of thanacytes must be due to an additional mechanism than solely the transient antibacterial role of phagocytic plasmatocytes. We thus tested whether some AMPs play a role in controlling S-*M. abscessus* infection.

The prediction was that the absence of such AMPs would result in rapid death of *Drosophila* already depleted of their phagocytic plasmatocytes.

To do so, we preinjected flies with clodronate liposomes (Clodrosomes), known to deplete phagocytic plasmatocytes (13). We used mutant flies defective for the AMPs-encoding genes (Fig. 1A) whose expression is either regulated by the Imd (group B, *AttC^Mi^*, *Dro-AttA-B^SK2^*, *Dpt^SK1^*, and *AttD^Sk1^*), the Toll pathway (bomanin, *Bom^Δ55C^*, and group C, *Mtk^R1^* and *Drs^R1^*), or both (*Def^SK3^* [group A] and *ΔCec^A-C^*) (14, 15). The controls for the Imd and the Toll pathways were *Relish* (*Rel^E20^*) and *spatzle* (*spz^rm7^*) mutant flies, respectively. Infected *ΔCec^A-C^*, group B, and *Bom^Δ55C^* mutant flies preinjected with Clodrosome were as susceptible as the control to S-*M. abscessus* infection (Fig. 1B, E, and G). In contrast, phagocyte-depleted flies of group A, *Rel*, *spz*, and group C were significantly more sensitive to the infection (Fig. 1C, D, F, and H), supporting that Defensin and Toll-regulated AMP genes (*Metchnikowin* and/or *Drosomycin*) may protect *Drosophila* from S-*M. abscessus* infection in the absence of phagocytic plasmatocytes or when the bacteria is extracellular.

**Increased survival of Tep4–opsonin-deficient flies infected by *M. abscessus*.** This first evidence leads us to a second hypothesis related to the propensity of S-*M. abscessus* to be found within phagocytic cells during infection. This advantage could be lost if the entry of S-*M. abscessus* into phagocytic cells is blocked.

Opsonization is a well-conserved process in multicellular organisms by which the infected host produces molecules that act as universal phagocytosis signals of invading pathogens, the opsonins. Indeed, binding of these proteins to the pathogen surface leads to its recognition by phagocytes. Examples of opsonins include complement factors in vertebrates (e.g., C3a) (16) or thioester-containing proteins (TEPs) in insects (17). Six Tep-encoding genes have been described in *Drosophila*, with *Tep5* not being expressed (18) and *Tep6* encoding a transmembrane protein devoid of thioester motifs (19). We thus decided to evaluate the potential implication of the four functional Teps (Tep1 to Tep4) as candidates required for internalizing S-*M. abscessus* into plasmatocytes, with the prediction that mutant flies for the implicated Teps would better survive S-*M. abscessus* infection due to the exposure of extracellular bacteria to AMPs.

First, we assessed the survival of infected flies which were homozygous mutants for a single gene (*Tep1*, *Tep3*, or *Tep4*), a combination (*Tep1* and *Tep4* or *Tep2* and *Tep3*), or the four genes (*Tepq^Δ^*) (20). The *Tepq^Δ^* mutant flies survived better than the control to S-*M. abscessus* infection. Among single or combinations of two *Tep* mutants, only flies defective at least for *Tep4* (*ΔTep4* or *Tep1*, *Tep4^Δ^* or *TEPq^Δ^*) were less sensitive to the infection than the control or other *Tep* mutants (Fig. 2A), suggesting that among *Drosophila*-inducible Teps, only Tep4 loss of function confers an increased survival against S-*M. abscessus* infection. We confirmed this observation with another homozygous *Tep4* mutant genotype (from the Bloomington collection) (see Fig. S1A in the supplemental material).

To exclude the possibility that the observed phenotype was caused by homozygosity for another mutation on the chromosome carrying *ΔTep4*, we used the *GAL4/UAS* system to express RNA interference (RNAi) to target *Tep4* transcripts in *Tep4*-expressing cells. mRNA depletion would at least mimic a hypomorphic *Tep4* mutant as confirmed by reverse transcription-quantitative PCR (qRT-PCR) (Fig. S1B). Consistently, 65% of: infected flies expressing *Tep4* RNAi in *Tep4* expression domain (*Tep4>Tep4* RNAi) were alive on day 10 dpi compared to only 20% for the *Tep4>* control flies (Fig. 2B). This result confirmed that the increased survival of *ΔTep4* flies was due to *Tep4* loss of function and raised the question the potential involvement of Tep4 in S-*M. abscessus* internalization.

Due to the technical impossibility of fully assessing bacterial internalization in adult *Drosophila*, we tested whether Tep4 might be involved in S-*M. abscessus* internalization using *Drosophila* S2 cells, an embryonic cell line which has hemocyte-like features. The latter cells have been used in the past to study cellular entry and restriction factors of mycobacteria (21–24). We first observed that S2 cells are permissive to S-*M. abscessus* infection as demonstrated by the intracellular growth over time (Fig. S1C). We next assessed Tep4 involvement in S-*M. abscessus* internalization by treating cells with *Tep4*-specific RNAi whose efficiency

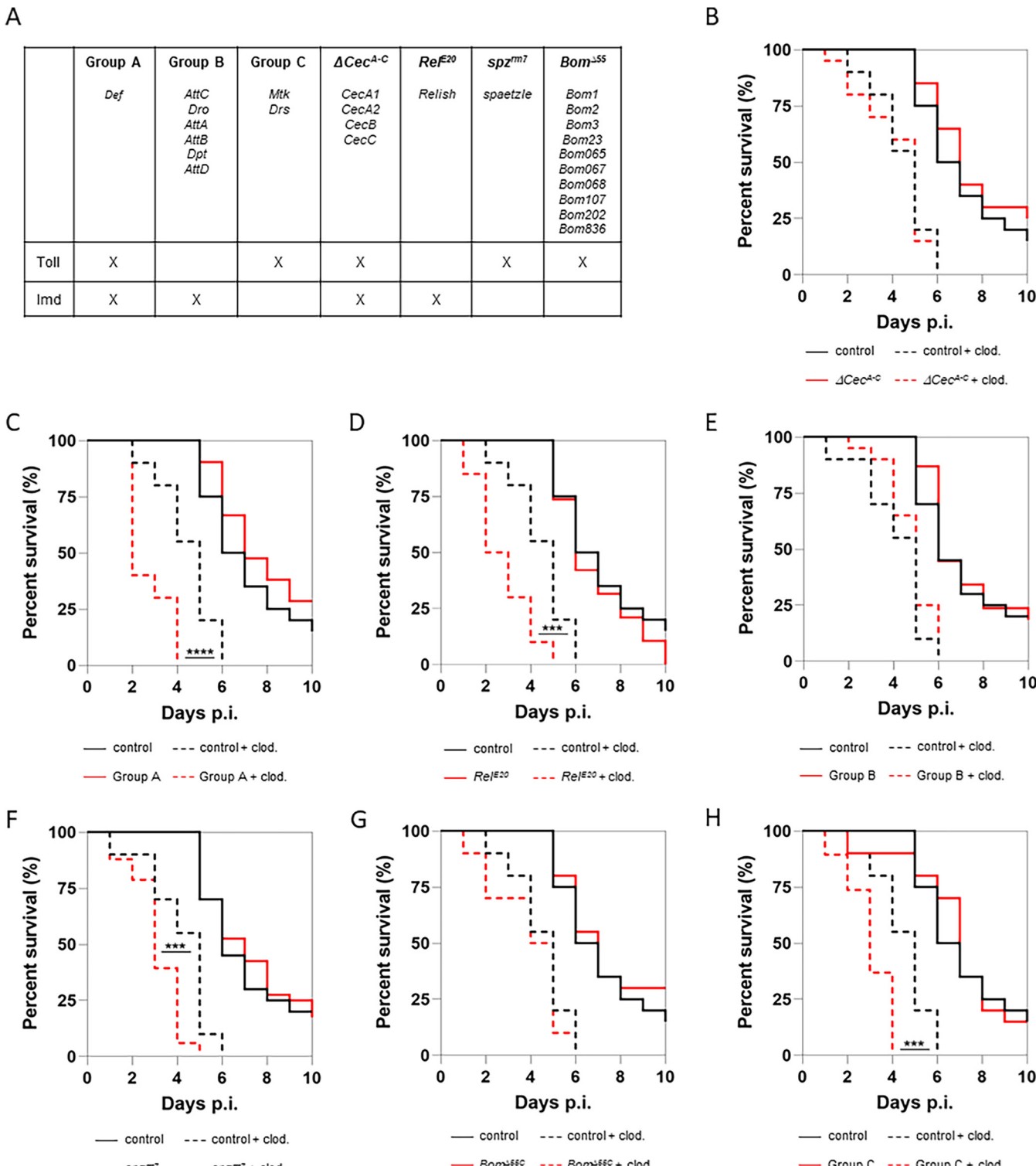

**FIG 1** AMPs are required for controlling extracellular *M. abscessus*. (A) Table summarizing the different losses of gene function by genotype (in bold at the top of the column). The X indicates which pathway (Toll or Imd) controls the production of AMPs (groups A to C and *Cec*[A-C]) or in which pathway the gene products are involved (*Rel*[E20], *spz*[rm7], and *Bom*[Δ55C]). (B) Survival curves of *w*[1118] (control) and cecropin-deficient (ΔCecA to ΔCecC) flies injected with water or/and beads and then injected with water or 10 CFU of *M. abscessus*. (C) Survival curves of *w*[1118] (control) and *Defensin*-deficient (group A) flies injected with water or/and beads and then injected with water or 10 CFU of *M. abscessus*. (D) Survival curves of *w*[1118] (control) and *Relish* mutant (*Rel*[E20]) flies injected with water or/and beads and then injected with water or 10 CFU of *M. abscessus*. (E) Survival curves of *w*[1118] (control) and *Drosocin*-, *Diptericin*-, and *Attacin-Deficient* (group B) flies injected with water or/and beads and then injected with water or 10 CFU of *M. abscessus*. (F) Survival curves of *w*[1118] (control) and *spaetzle*-deficient (*spzrm7*) flies injected with water or/and beads and then injected with water or 10 CFU of *M. abscessus*. (G) Survival curves of *w*[1118] (control) and *Bomanin*-deficient (*Bom*[Δ55C]) flies injected with water or/and beads and then injected with water or 10 CFU of *M. abscessus*. (H) Survival curves of *w*[1118] (control) and *Drosomycin*- and *Metchnikowin-Deficient* (group C) flies injected with water or 10 CFU of *M. abscessus*. Survival was analyzed on 40 to 60 flies per condition using the log-rank test (***, $P < 0.001$; ****, $P < 0.0001$).

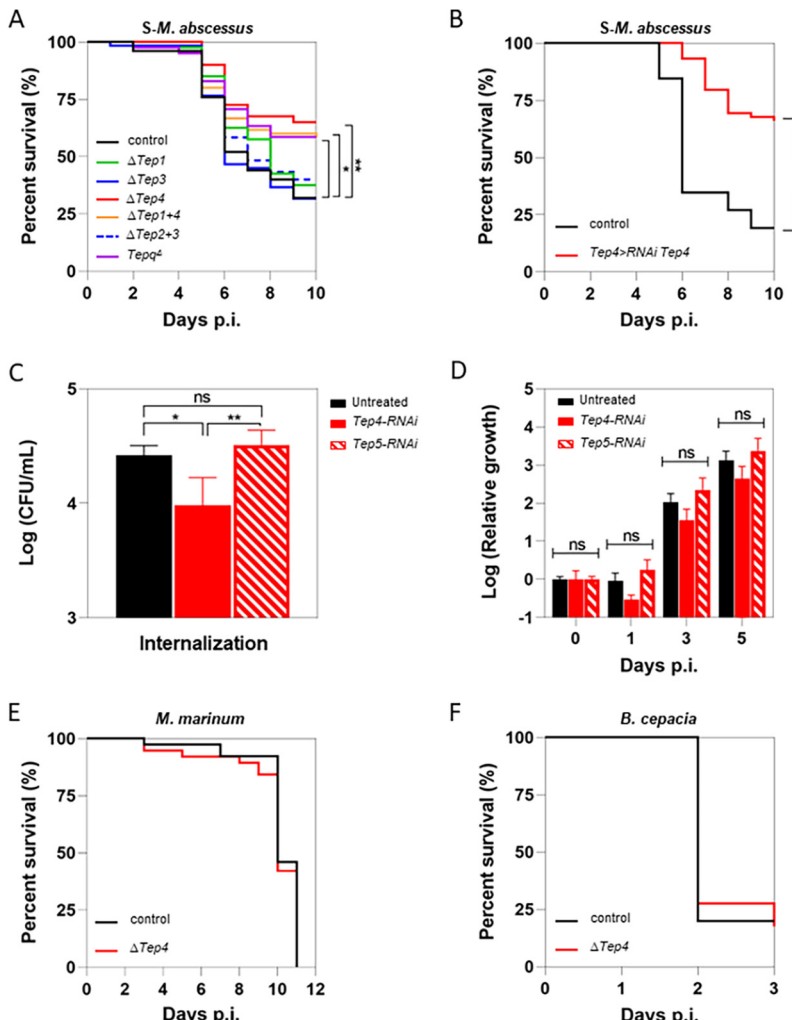

**FIG 2** *Tep4*-deficient flies have an increased survival against *M. abscessus* infection. (A) Survival curves of *w^1118* (control) or mutant flies for *Tep1* (Δ*Tep1*), *Tep3* (Δ*Tep3*), *Tep4* (Δ*Tep4*), *Tep1* and *Tep4* (Δ*Tep1+4*), *Tep2* and *Tep3* (Δ*Tep2+3*), or *Tep1*, *Tep2*, *Tep3*, and *Tep4* (*Tepq^Δ*) injected with 10 CFU of *M. abscessus*. (B) Survival curves of *w^1118* (control) and *Tep4>*RNAi *Tep4* flies injected with 10 CFU of *M. abscessus*. (C) Internalization of *M. abscessus* by untreated S2 cells, with *Tep4-* or *Tep5*-specific dsRNA-treated conditions. Bacterial load was quantified by CFU counting. (D) Relative intracellular bacterial growth over the time according to day 0 of untreated or *Tep4-* or *Tep5*-specific dsRNA-treated S2 cells. (E) Survival curves of *w^1118* (control) and *Tep4* mutant (Δ*Tep4*) flies injected with 10 CFU of *M. marinum*. (F) Survival curves of *w^1118* (control) and *Tep4* mutant (Δ*Tep4*) flies injected with 10 CFU of *B. cepacia*. CFU were counted in 3 independent experiments, on 5 × 10⁴ cells infected at MOI of 1:10. One-way ANOVA and two-way ANOVA analysis were performed in panels A and B, respectively (*, $P < 0.05$; **, $P < 0.01$). Survival was analyzed for 40 to 60 flies per condition using the log-rank test (***, $P < 0.001$; ****, $P < 0.0001$). p.i., postinfection.

was confirmed (Fig. S1D). *Tep4*-RNAi-treated cells internalized less S-*M. abscessus* than untreated cells (Fig. 2C), suggesting that Tep4 is required for *M. abscessus* internalization. As a control, cells treated with an RNAi targeting the transcripts of the *Tep5* pseudogene displayed a similar internalization to untreated cells (Fig. 2C). Kinetic measurement of the relative growth of S-*M. abscessus* in S2 cells compared to day 0 revealed no significant difference between *Tep4*-RNAi-treated cells and control (Fig. 2D). These results support that Tep4 is required for S-*M. abscessus* internalization and not subsequent steps once the bacteria are internalized.

It has been previously reported that *Tep4* mutant flies are as susceptible as wild-type control to infection with 500 CFU of *Mycobacterium marinum* (19), a strict pathogenic nontuberculous *mycobacterium* that also grows in *Drosophila* hemocytes (25). We confirmed this observation by injecting 10 CFU of *M. marinum* (Fig. 2E). Similarly, Δ*Tep4* flies injected with 10 CFU of *Burkholderia cepacia*, a Gram-negative and extracellular bacterium, had similar

survival to wild-type control flies (Fig. 2F), suggesting that the increased survival of Tep4-defective flies is a feature of S-*M. abscessus* infection and not an intrinsic resistance of these flies to any bacterial infection.

In conclusion, we observe that in the absence of Tep4, flies have better survival after infection with *M. abscessus*, but not with *M. marinum*. Knowing the role of Tep4 as an opsonin, we therefore emitted the hypothesis of a predominantly extracellular presence of *M. abscessus* as for *M. marinum*, but where only *M. abscessus* was controlled. We then hypothesized that the increased survival of Δ*Tep4* mutant flies during *M. abscessus* infection could be due to a better control of the infection through AMP overproduction, and this is what we tried to elucidate in the rest of this work.

**Induction of AMP transcripts in Tep4-depleted flies during *M. abscessus* infection.** The increased survival of Δ*Tep4* flies indicated better control of *M. abscessus* infection, confirmed by a lower mycobacterial load on 3 dpi than the control (Fig. 3A).

This increase of extracellular S-*M. abscessus* might activate the host humoral immune response. Thus, we measured, by qRT-PCR, the transcript levels of AMP-encoding genes in the two humoral response pathways in *Drosophila* mutated in Δ*Tep4* flies and compared them to that observed in wild-type flies (Fig. 3A). The transcript levels of the main AMP-encoding genes were significantly increased in infected *Tep4* mutants for almost all tested AMP-encoding genes, whether regulated by both NF-$\kappa$B (Fig. 3B and C), Imd (Fig. 3D and F), or Toll (Fig. 3H) pathways. The surprising exceptions were *Diptericin* and *Drosomycin* genes, whose transcript levels decreased (Fig. 3E) or did not change (Fig. 3G), respectively. In parallel, we measured the levels of transcripts of peptidoglycan recognition protein-SB1 (*PGRP-SB1*) and *PGRP-SD*, encoding two of the main equivalents of bacterium-specific pattern recognition receptors (PRR) in *Drosophila* (26). The transcript levels of both genes were increased in S-*M. abscessus*-infected *Tep4* mutant flies compared to the control (Fig. S2A and B), suggesting that bacterial-sensing factors are upregulated in infected Δ*Tep4* flies.

Taken together, these results supported that the better control of *M. abscessus* infection by Δ*Tep4* flies would be related to increased production of AMPs during infection. Two additional implications based on these different results are that the presence of Tep4 would be deleterious to the fly in two possible ways, by promoting *M. abscessus* internalization into phagocytic cells and by reducing the AMP response during infection, as shown here.

To assess a protective role for AMPs in the absence of Tep4, we infected Tep4-depleted flies that were either heterozygous for group A (*Def^SK3^*) or group C mutants. These mutants were selected based on the observations made in flies preinjected with Clodrosome. We observed in these flies an increased susceptibility to S-*M. abscessus* infection, mainly for group A (*Def^SK3^*), compared to the *M. abscessus*-infected *Tep4>Tep4* RNAi flies (Fig. 3I and J). Defensin seems to be a major factor in the humoral response against *M. abscessus* infection, as shown by the consequences of its absence in flies depleted of their phagocytes (Fig. 1B) and its depletion combined with that of *Tep4* transcripts (Fig. 3I). This raises the question of the role of defensin in the control of *M. abscessus* infection in Tep4-depleted flies.

**Defensin controls *M. abscessus* infection in *Drosophila*.** We first assessed whether Defensin was sufficient to control *M. abscessus* infection *in vivo*. Infected flies overexpressing constitutively and ubiquitously *Defensin* (*Tub>Defensin*) were significantly less susceptible to the infection than the wild-type control (Fig. 4A). We validated the upregulation of defensin in *Tub>Defensin* flies (Fig. S3). In comparison, using a previously validated transgene allowing *Drosocin* overexpression (27), similar survival rates were observed between control flies and flies overexpressing *Drosocin* (Fig. 4B). This last gene was chosen as a negative control because its absence (group B mutant) in phagocyte-depleted flies (Fig. 1D) did not have any impact.

We then tested whether Defensin could inhibit the growth of S-*M. abscessus*. To do so, we cultured S-*M. abscessus* in the presence or absence of physiological doses of synthetic Defensin. Addition of increased concentrations of synthetic defensin in the culture media led to an inversely proportional fluorescent signal of resazurin with both 100 and 1000 CFU of initial inoculum (Fig. 4C). These results indicate that Defensin is capable of killing *M. abscessus in vitro*. Amikacin, an antibiotic known to kill *M. abscessus*, was used as the

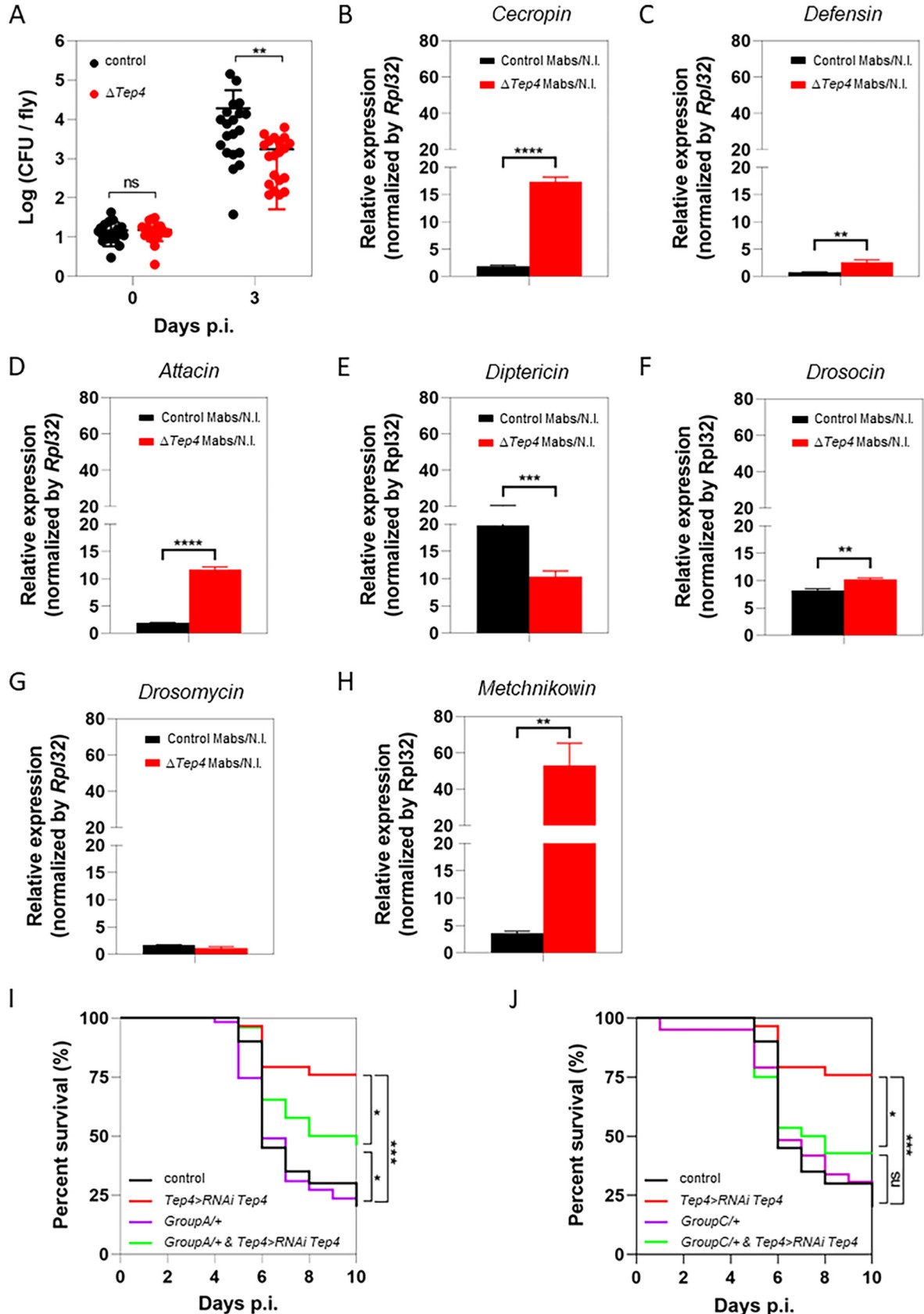

**FIG 3** *Tep4*-deficient flies' AMP overexpression increases their survival against *M. abscessus*. (A) *M. abscessus* bacterial load quantification in *w^1118^* (control) and *Tep4*-deficient flies (Δ*Tep4*) on days 0 and 3 postinfection with 10 CFU of *M. abscessus*. (B to H) Quantification of

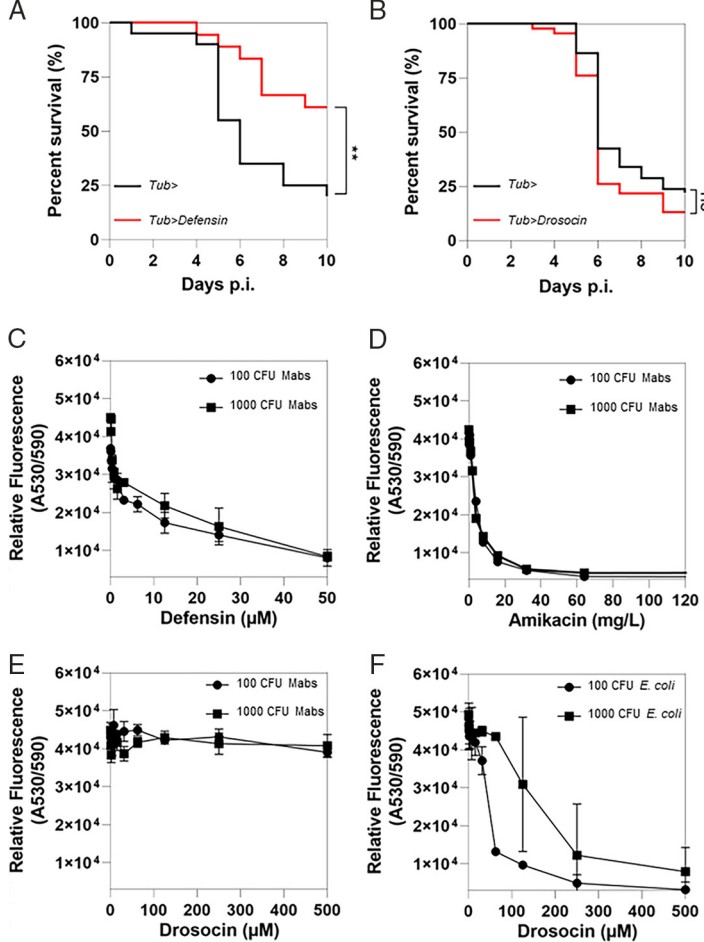

**FIG 4** Defensin is sufficient to control *M. abscessus* infection through a direct bactericidal action. (A) Survival curves of *Tubulin>* (Tub>) and *Tubulin>Defensin* (Tub>*Defensin*) flies injected with 10 CFU of *M. abscessus*. (B) Survival curves of *Tub>* and *Tubulin>Drosocin* (Tub>*Drosocin*) flies injected with 10 CFU of *M. abscessus*. (C to F) Assessment of microbial growth inhibition by reading resazurin fluorescence intensity using a spectrophotometer with excitation wavelength at 385 nm and emission at 460 nm under the conditions of addition of *Defensin* to *M. abscessus* in culture (C), addition of amikacin to *M. abscessus* in culture (D), addition of Drosocin to *M. abscessus* in culture (E), and addition of Drosocin to *E. coli* in culture (F). Survival was analyzed for 40 to 60 flies per condition using the log-rank test (**, $P < 0.01$).

positive control regardless of the inoculum (Fig. 4D). No bacterial killing was observed when S-*M. abscessus* was cultured in the presence of synthetic Drosocin (Fig. 4E) that was able, by comparison, to inhibit the growth of *Escherichia coli* (28) (Fig. 4F).

Collectively, these results demonstrate that the direct bactericidal activity of Defensin plays a major role in *Drosophila* humoral response to S-*M. abscessus* infection.

## DISCUSSION

*M. abscessus* causes pulmonary and mucocutaneous infections in healthy and immuno-compromised persons (29). The increased prevalence of *M. abscessus* infections, especially in

**FIG 3 Legend (Continued)**

AMP-encoding gene relative expression by qRT-PCR. RNA was extracted on day 3 postinfection from $w^{1118}$ and Δ*Tep4* flies injected with water or 10 CFU of *M. abscessus*. Plots represent the ratio of expression levels in infected on noninfected flies for *Cecropin C* (B), *Defensin* (C), *Attacin A* (D), *Diptericin* (E), *Drosocin* (F), *Drosomycin* (G), and *Metchnikowin* (H). (I) Survival curves of $w^{1118}$ (control), *Tep4>*RNAi Tep4, defensin-deficient (groupA/+), and Tep4>RNAi Tep4 heterozygous for the defensin mutation (group A/+ and *Tep4>*RNAi *Tep4*) flies injected with 10 CFU of *M. abscessus*. Survival was analyzed for 40 to 60 flies per condition using the log-rank test (*, $P < 0.05$; ***, $P < 0.001$). Bacterial loads were individually quantified from 6 flies per condition. Bacterial load and AMP-encoding gene expression data were analyzed using a two-way ANOVA statistical test. (*, $P = 0.05$; **, $P < 0.01$; ***, $P < 0.001$; ****, $P < 0.0001$). (J) Survival curves of $w^{1118}$ (control), *Tep4>*RNAi *Tep4*, *Drosomycin*- and *Metchnikowin*-deficient (group C/+), and *Tep4>*RNAi *Tep4* heterozygous for *Drosomycin*- and *Metchnikowin*-deficient (group C/+ and *Tep4>*RNAi *Tep4*) flies injected with 10 CFU of *M. abscessus*. Survival was analyzed for 40 to 60 flies per condition using the log-rank test (*, $P < 0.05$; ***, $P < 0.001$).

patients with cystic fibrosis, drives the need to understand the infection process and mechanisms of resistance of the mycobacteria to the host immune system. To date, *Drosophila* remains the main model susceptible to infection with the S morphotype, the infective form of this *mycobacterium* (30, 31). The innate response of fruit flies is based on a cellular and humoral response, the latter mainly relying on AMP production mediated by the Toll and Imd pathways (11).

We recently described the crucial role of cellular responses in controlling S-*M. abscessus* infection. Indeed, after systemic injection, S-*M. abscessus* was rapidly internalized by *Drosophila* phagocytes (8). In parallel, the humoral response was also activated in response to the infection. Despite the induction of AMP-encoding gene expression, mutant flies for these genes had survival rates similar to those of wild-type flies after S-*M. abscessus* injection. We hypothesized that (i) AMPs would be ineffective against this *mycobacterium*, and/or (ii) the rapid internalization of S-*M. abscessus* would protect it from the bactericidal action of AMPs.

Our results obtained by depleting phagocytes in AMP-deficient flies or by limiting mycobacterial entry into phagocytes validate the second hypothesis and support that Defensin, Drosomycin, and Metchnikowin were necessary to fight against extracellular S-*M. abscessus in vivo*. Ultimately, cultures of S-*M. abscessus* in the presence of physiological doses of AMPs showed that defensin kills this *mycobacterium* through direct bactericidal action, confirming the results obtained by their overexpression *in vivo*.

This direct killing of *M. abscessus*, an RGM, is reminiscent of the activity of some mammalian beta defensins on some slow-growing mycobacteria (i.e., beta defensin 1 on *M. tuberculosis* [32, 33] and beta defensin 5 on *M. bovis* [34]). However, *Drosophila* defensin belongs to the *cis*-defensins group, while the beta defensins belong to the *trans* group (35, 36). Whether these two superfamilies have a common phylogenetic origin or are a product of a convergent evolution is still under debate (37).

Defensin is added to the panoply of peptides produced by arthropods that are active against some RGM. Indeed, peptides extracted from venoms of some Brazilian wasps (*Polybia dimorpha* [38], *Pseudopolybia vespiceps testacea* [39]) or scorpions (*Hadrurus gertschi* [40], *Tityus obscurus* [41]) are active on another member of the *M. abscessus* complex, *M. massiliense*, *in vitro* and *ex vivo*. Importantly, they are able to kill this *mycobacterium* inside mammalian macrophages in culture (38–41) as beta defensin 1 on *M. tuberculosis* (32, 33). We have not directly assessed whether Defensin is also able to penetrate mammalian phagocytes or other cell types, but if not, it might be possible to deliver it with nanoparticles as used for bovine beta defensin 5 against *M. bovis* (34).

In the case of a systemic infection, the Toll pathway activates the expression of *Drosomycin* and *Metchnikowin* and partially that of *Defensin* (42–44). Interestingly, Oh et al. reported increased susceptibility in flies defective for this pathway (*PGRP-SA* and *Dif*), infected by injecting 4,000 CFU of S-*M. abscessus* (30). We did not observe this by injecting 10 CFU to mutant flies for *spz*, which encodes another component of this NF-$\kappa$B pathway. This discrepancy could be explained by the difference in the number of mycobacteria used in the two studies. After injection of a large amount of *M. abscessus*, one part would be internalized, with the other part remaining extracellular. The consequence would be an activation of the Toll pathway, and thus, the induction of the production of Drosomycin, Metchnikowin, and Defensin, whose importance against *M. abscessus* is revealed by our study.

Consistent with this previous report (30), the Imd pathway does not seem to be required to fight *M. abscessus* infection in wild-type flies. We also observed that *rel* mutant flies became more susceptible to mycobacterial infection when their phagocytes were depleted. However, phagocyte depletion in a mutant fly for Attacin and Drosocin, whose expression is regulated by the Imd pathway, did not result in increased susceptibility to *M. abscessus* infection, suggesting that these AMPs were not required in the response to the presence of extracellular mycobacteria mediated by this NF-$\kappa$B pathway. With *Defensin* expression also activated by this pathway (45), this AMP is a good candidate for mediating the Imd response in this context.

We confirmed the protective role played by *M. abscessus* internalization, and by limiting mycobacteria entry into phagocytes such as in *Tep4*-deficient flies, we exposed *M. abscessus*

to action of AMPs. We found that *Tep4*-deficient flies' increased resistance was dependent on *Defensin* and *Metchnikowin* overexpression. This is consistent with the observations of increased *Defensin* and *Cec-A1* expression in *Tep4* mutant flies compared to wild-type flies after infection with intracellular *Photorhabdus* sp. (46). Depletion of defensin and other AMPs in this context will address whether this resistance to these infections is dependent on the AMPs production, as we reported here. In accordance with this previous study of the *Tep4* mutant, we also found no increased expression of these two genes in uninfected *Tep4* mutant flies (46) and extended it to other AMP-encoding genes, adding support for the hypothesis that microbes injected in the hemolymph are responsible for AMP-encoding gene expression.

In insects, Tep proteins are equivalent to complement factors in vertebrates (47) and act as opsonins (48). Among them, the most studied is *Anopheles gambiae* Tep1, which is phylogenetically related to the complement factor C3 (49) and opsonizes microbes in a thioester-dependent manner (48). Six Tep-encoding genes have been described in *Drosophila*. *Tep5* does not appear to be expressed (18), and the product of *Tep6* is nonfunctional (19). Among the four inducible Teps (Tep1 to Tep4), Tep2 is required for efficient phagocytosis of the Gram-negative *E. coli* but not of the Gram-positive *Staphylococcus aureus*. Conversely, Tep3 is required for the phagocytosis of *S. aureus* but not *E. coli* (50). This indicates the microbial specificity of Tep proteins to act as opsonins. It also seems to be the case for Tep4. Indeed, this protein acts as an opsonin for the internalization of the extracellular bacteria *S. aureus* and *Enterococcus faecalis* (20), the facultative intracellular bacterium *Pseudomonas aeruginosa* (19, 51), the intracellular *Photorhabdus* sp. (52), and now *M. abscessus*. We confirm the previous observation that Tep4-deficient flies were as sensitive as wild-type flies during a challenge with *M. marinum*, a slow-growing, strictly pathogenic *mycobacterium* (19). This could be explained by the fact that flies infected by *M. marinum* fail to elicit the production of AMPs (25). Nonmutually exclusive alternative explanations are (i) AMPs would not be effective against *M. marinum*, even when this *mycobacterium* is extracellular; and (ii) internalization of these two mycobacteria would not require the same opsonins; either Tep4 would not be a specific opsonin of *M. marinum* or its absence would be compensated, in contrast to S-*M. abscessus*. Although specific factors for mycobacterial entry into phagocytes have been identified, such as Peste (22), further studies are required to identify the recognition and interaction of host elements specific to mycobacterial species.

## MATERIALS AND METHODS

**Bacterial strains and cultures.** The following microbial strains were used: *M. abscessus* subsp. *abscessus* (*M. abscessus* here; ATCC 19977), *M. marinum* wild-type M (collection of Institut Pasteur 64.23), *Escherichia coli* (DH5 strain), *Burkholderia cepacia* (clinical strain), and *Candida albicans* (clinical strain). All strains were grown at 37°C except *M. marinum* (28°C). *M. abscessus* and *M. marinum* were grown in Middlebrook 7H9 medium (Sigma-Aldrich, St. Louis, MO, USA) supplemented with 1% glucose and glycerol 0.2% at aerobic condition until an optical density at 600 nm ($OD_{600}$) between 0.6 and 0.8 was reached. *E. coli*, *B. cepacia*, and *C. albicans* were cultured in standard Luria-Bertani (LB) medium. Bacterial cultures were then centrifuged to get concentrated aliquots that were then frozen at −80°C in 10% glycerol.

***Drosophila* maintenance, crosses, and infection.** Flies were raised on standard corn agar medium at 25°C. Crosses were performed at 25°C. Transgene expression was performed using the *UAS-GAL4* system (53). The following stocks obtained from the Bloomington Drosophila Stock Center (NIH P40OD018537) were used in this study: Δ*Tep4* (*y*[1] *w*[67c23]; *P*{*y*[1mDint2] *w*[1mC]=*EPgy2*}*Tep4*[EY04656]; stock no. 15936), *Tep4-Gal4* (*y*[1] *w*[*]; *Mi*{*Trojan-GAL4.2*}*Tep4*[MI13472-TG4.2]/*SM6a*; stock no. 76750); *UAS Tep4-RNAi* (*y*[1] *sc*[*] *v*[1] *sev*[21]; *P*{*y*[1t7.7] *v*[1t1.8]=*TRiP.GL00395*}*attP2*; stock no. 35469). The following *Drosophila* strains lacking antimicrobial peptide (AMP) production were obtained from B. Lemaitre (14, 15): DrosDel *iso w*[1118] (reference strain), *iso* *Bom*[Δ55C] (Toll pathway mutant), *iso Rel*[E20] (Imd pathway mutant), *iso Spz*[rm7] (Toll pathway mutant), group A (defensin mutant *iso*; *Def*[SK3]), group B (mutants for the IMD-related drosocin, diptericins, and attacins, *iso*, *AttC*[Mi], *Dro-AttA-B*[SK2], *DptA-B*[Ski], and *AttD*[Ski]), group C (mutants for the Toll-related metchnikowin and drosomycin, *iso*, *Mtk*[R1], and *Drs*[R1]) and Δ*Cec*[A-C] (mutant for *Cecropin A1, A2, B*, and *C*). The following mutant for *Tep* genes was obtained from B. Lemaitre: *Tepq*[Δ], Δ*Tep1*, Δ*Tep3*, Δ*Tep4*, Δ*Tep1+4*, and Δ*Tep2+3* (20). F. Rouyer provided us *w*[1118] (from NIG-Fly), J. P. Parvy provided us *UAS-Defensin* flies, and S. Chtarbanova provided the *UAS-Drosocin* ones.

Frozen bacterial aliquots were thawed on ice and homogenized with a 30-gauge insulin needle (Becton, Dickinson, France) to avoid clumps. Serial 10-fold dilutions were done, and 30 μL of each dilution was spread on a blood agar plate for mycobacteria (Columbia agar with 5% sheep blood [COS]; bioMérieux, France) or on a classic LB agar plate for *B. cepacia*. Plates were then stored at 37°C for 2 or 3 to 7 days depending on bacteria, and CFU counts were determined.

**TABLE 1** List of primers used in this study

| Primer | Sequence (5′–3′) |
| --- | --- |
| Oligonucleotides used for qPCR | |
| *PGRP-SB1* forward | TTAGCTCTATCCGCCAATGC |
| *PGRP-SB1* reverse | CCCTTGTGATCCGACTGAAT |
| *PGRP-SD* forward | ATGACTTGGATCGGTTTGCT |
| *PGRP-SD* reverse | GCTGGGAGCATGTAACATCA |
| *Tep4* forward | GCTGCAGAACCAGATCGAAATC |
| *Tep4* reverse | ATGACTTTGGCGACGTCTTGAT |
| *Attacin A* forward | CGTTTGGATCTGACCAAGG |
| *Attacin A* reverse | AAAGTTCCGCCAGGTGTGAC |
| *Cecropin C* forward | TCATCCTGGCCATCAGCATT |
| *Cecropin C* reverse | CGCAATTCCCAGTCCTTGAAT |
| *Defensin* forward | GAGGATCATGTCCTGGTGCAT |
| *Defensin* reverse | TCGCTTCTGGCGGCTATG |
| *Diptericin A* forward | GCGGCGATGGTTTTGG |
| *Diptericin A* reverse | CGCTGGTCCACACCTTCTG |
| *Drosocin* forward | TTTGTCCACCACTCCAAGCAC |
| *Drosocin* reverse | ATGGCAGCTTGAGTCAGGTGA |
| *Drosomycin* forward | CTGCCTGTCCGGAAGATACAA |
| *Drosomycin* reverse | TCCCTCCTCCTTGCACACA |
| *Metchnikowin* forward | AACTTAATCTTGGAGCGATTTTTCTG |
| *Metchnikowin* reverse | ACGGCCTCGTATCGAAAATG |
| *RpL32* forward | AGCATACAGGCCCAAGATCG |
| *RpL32* reverse | TGTTGTCGATACCCTTGGGC |
| Oligonucleotides used for PCR template for dsRNA synthesis | |
| *Tep4* forward | TAATACGACTCACTATAGGGGACCTGGATCTTTGCCGATA |
| *Tep4* reverse | TAATACGACTCACTATAGGGTTCCAGCTCGAAGGTGTTCT |
| *Tep5* forward | TAATACGACTCACTATAGGGCCGGAAAAAGGATTGGGTAT |
| *Tep5* reverse | TAATACGACTCACTATAGGGGACCAGTCTGGAGTTTGGGA |

The bacterial inoculum was diluted in water to get suitable concentrations. Five- to 7-day-old virgin female flies were anesthetized with $CO_2$ (Inject-Matic, Switzerland) and were infected with 50 nL of the suspension containing 10 bacteria by injection into the sternopleural suture. Infections were performed using a nano-injector Nanoject III (Drummond Scientific Company, USA) charged with a calibrated pulled glass needle made with a DMZ Universal electrode puller (Zeitz Instruments, Germany). To chemically deplete the phagocytes, 24 h prior to infection with *M. abscessus*, flies were individually preinjected with 69 nL of clodronate liposomes (Clodrosome; CLD-8901; Encapsula, USA) diluted in phosphate-buffered saline (PBS; ratio of 1:5). Flies were kept anesthetized no more than 10 min. Infected flies were maintained at 28°C under controlled humidity conditions. Twenty flies were used for every experimental condition, and each experiment was performed at least in three independent replicates. Mortality was registered daily and surviving flies were transferred into a new vial every 2 days, until 10 dpi except for *M. marinum* (12 dpi).

**Bacterial load quantification.** Twenty infected flies per experimental condition were individually grounded in 250 $\mu$L of water using sterile polypropylene cones (Kimble 749521-1590; Kimble Chase, USA). The broths were centrifuged at 1,200 × *g* for 2 min and diluted by 10-fold serial dilutions. We spread 50 $\mu$L of each dilution on VCA3 plates (VCA3; bioMérieux, France) containing selective antibiotics for *M. abscessus* (vancomycin, colistin, trimethoprim, and amphotericin B). The plates were kept at 37°C for 1 week.

**S2 cell maintenance, treatment with dsRNA, infection, and assessment of *M. abscessus* internalization and growth.** S2 cells were cultured in complete Schneider medium (Gibco; catalog no. 11590576) at 25°C without $CO_2$. *Tep4*- and *Tep5*-specific double-stranded RNA (dsRNA) was produced and purified with the MegaScript T7 kit (Invitrogen; catalog no. AM1333) following the manufacturer's instructions, using PCR products from S2 cell DNA as the template. The primers used for the PCR are listed in Table 1. For each condition, $10^6$ cells were treated with 20 $\mu$g of dsRNA and maintained in culture for 4 days, as described in reference 54. For each condition, $5 \times 10^4$ cells were infected with *M. abscessus* at a multiplicity of infection of 1:10. Internalization was assessed by infecting cells for 1 h at 28°C; cells were then washed 3 times with 1× PBS and lysed in water. The number of internalized bacteria was determined by CFU counting. For the assessment of *M. abscessus* growth in S2, cells were infected for 3 h at 28°C, washed 3 times with 1× PBS, and incubated for 1 h with amikacin at 250 mg/L to kill extracellular bacteria. Cells were then washed 3 times with 1× PBS and incubated at 28°C with 1 $\mu$g of dsRNA when needed and amikacin at 50 mg/L. Cells were washed 3 times with 1× PBS and lysed in water on 0, 1, 3, and 5 dpi, and bacterial load was determined by CFU counting.

**Assessment of microbial growth inhibition by AMPs and amikacin.** One hundred or 1,000 CFU of *M. abscessus*, *E. coli*, or *C. albicans* were grown in 200 $\mu$L of Mueller-Hinton medium. The culture media contained 0.125 to 500 $\mu$M synthetic *Drosophila* defensin or drosocin (Genepep, Saint-Jean-de-Védas, France) obtained by 2-fold serial dilution in a 96-well plate. To quantify bacterial growth, 20 $\mu$L of resazurin (60 $\mu$g/mL) was added in each well on day 4 after inoculation for *M. abscessus* and day 1 for *E. coli*, and resazurin fluorescence

was quantified using a spectrophotometer (Tecan Infinite M200; Life Sciences) with excitation wavelength at 385 nm and emission at 460 nm. Three independent experiments were performed for each condition.

**qRT-PCR.** Total RNA was extracted from 20 female flies per condition using TRIzol reagent (Thermo Fisher, Waltham, USA), chloroform, and isopropanol. Genomic DNA was removed from the extracted RNA using Turbo DNA-free kit (catalog no. AM1907, Invitrogen, USA), and cDNA was generated using Superscript III (catalog no. 18080051; Invitrogen, USA) following the manufacturer's instructions. qPCR was performed in Maxima SYBR Green master mix (catalog no. K0221; Thermo Fisher, USA), using 100 ng of cDNA as the template and 10 $\mu$M target gene-specific primers. qPCR was performed with CF96 Touch real-time PCR detection system (Bio-Rad). The primers used are presented in Table 1. *RpL32* transcript levels were used for normalization.

**Biostatistical analysis.** All data were analyzed by GraphPad Prism 9.0.0 (GraphPad Software Inc., USA). The log-rank (Mantel-Cox) test for Kaplan-Meier survival curves was used to evaluate survival statistical significance. CFU quantification and gene expression levels were compared by one-way and two-way analysis of variance (ANOVA) when needed. $P$ values of $<0.05$ were considered significant.

**Data availability.** All relevant data are within the manuscript and its supplemental material files.

## SUPPLEMENTAL MATERIAL

Supplemental material is available online only.

**SUPPLEMENTAL FILE 1**, PDF file, 0.2 MB.

## ACKNOWLEDGMENTS

We thank B. Marshall (Southampton University Hospitals Trust, UK) for careful review of the manuscript. We also thank S. Alé-Lohème, J. M. Corsi, and V. Le Moigne for their assistance in the laboratory and N. Doisne and E. Balse (UMR-S 1166) for forging the capillaries for microinjections. We thank M. Crozatier, B. Lemaitre, J. P. Parvy. and S. Chtarbanova for generously giving us fly lines and the Bloomington *Drosophila* Stock Center (BDSC) for fly stocks.

H.T., J.-L.H., F.G.-M., and S.S. designed the experiments. H.T. performed all the experiments. N.D. helped in setting up RNAi production. H.T., J.-L.H., F.G.-M., and S.S. analyzed the data. H.T., J.-L.H., and S.S. wrote the manuscript. N.D., I.G., and F.G.-M. reviewed the manuscript.

We declare no competing interests.

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
