## [Reviewer comments · Microbiology Spectrum]

Microbiology Spectrum

Mycobacterium abscessus* opsonization allows an escape from the Defensin bactericidal action in *Drosophila

Hamadoun Touré, Nicolas Durand, Isabelle GUENAL, Jean-Louis Herrmann, Fabienne Girard-misguich, and Sébastien SZUPLEWSKI

Corresponding Author(s): Fabienne Girard-misguich, Université de Versailles Saint-Quentin-en-Yvelines

Review Timeline:

Submission Date:	February 20, 2023
Editorial Decision:	March 24, 2023
Revision Received:	April 20, 2023
Accepted:	May 5, 2023

Editor: Olivier Neyrolles

Reviewer(s): Disclosure of reviewer identity is with reference to reviewer comments included in decision letter(s). The following individuals involved in review of your submission have agreed to reveal their identity: Patricia M Lenhart-Pendergrass (Reviewer #1); Jichan Jang (Reviewer #2)

Transaction Report:

DOI: <https://doi.org/10.1128/spectrum.00777-23>

March 24, 2023

Dr. Fabienne Girard-misguich
Universite de Versailles Saint-Quentin-en-Yvelines
Versailles
France

Re: Spectrum00777-23 (*Mycobacterium abscessus* opsonization allows an escape from the Defensin bactericidal action in *Drosophila*)

Dear Dr. Fabienne Girard-misguich:

Thank you for submitting your manuscript to Microbiology Spectrum.

Your manuscript has been evaluated by two reviewers. Although the reviewers were generally enthusiastic about the manuscript, they raised several issues that need to be resolved before we can proceed further in the editorial process. In addition, they both noted that the manuscript is largely based on another study (Touré et al. In revision) which they would like to have access to and which they think should be published before the present study. I urge you to be scrupulous in meeting this expectation.

Link Not Available

Sincerely,

Olivier Neyrolles, PhD
Editor, Microbiology Spectrum

Journals Department
Reviewer comments:

Reviewer #1 (Comments for the Author):

In this study, the authors investigate the role of antimicrobial peptides and the opsonin Tep4 in control of *M. abscessus* (Mab) infection in *Drosophila*. This manuscript builds upon the findings of a prior manuscript that is evidently in revision at present,

which per the authors' description, demonstrated that Mab is internalized by *Drosophila* phagocytic plasmacytes, but was able to survive lysis of the phagocytic plasmacytes by NK cell-like thanocytes. *Drosophila* were resistant to Mab infection when thanocytes were depleted. Furthermore, they report that their previous work showed that AMPs are induced by Mab infection in *Drosophila*. In this study, they looked to build upon this finding by investigating the role of AMPs in Mab infection, and how Mab may evade the AMP response in *Drosophila*. The authors find that opsonization with Tep4 allows internalization of Mab which may allow it to evade host AMPs.

Major critiques

- Lines 133 - 134 - A major aspect of the conclusions of this manuscript seem to rest upon the concept that the Mab are extracellular when the phagocytic plasmacytes are depleted, however, this is not actually shown. Are there other cell types that could be susceptible to intracellular Mab infection in the absence of the phagocytic plasmacytes? This should be addressed as it's unclear if it already was addressed in the previous under revision manuscript or not.
- Figure 2A - the use of CFUs alone is not adequate to definitively state that Tep4 is involved in internalization. The experimental design would be unable to distinguish between internalized Mab versus Mab bound to the outer surface of the cell line. A complementary method such as flow cytometry or confocal microscopy with quenching should be utilized to make this distinction.
- Figure 2- the manuscript lacks evidence for the efficacy of the Tep4 RNAi - this should be shown in a figure or at a minimum, should be included in the text of the results section.
- Figure 2 - it is stated that Tep4 has been shown to act as an opsonin in the literature but it isn't directly demonstrated here. Results would be strengthened with addition of a western blot showing that Tep4 directly binds to Mab in the *Drosophila* model.
- Figure 3 and discussion - what is the proposed mechanism by which Tep4 depletion upregulates antimicrobial peptide expression?
- Figure 3B-D - it appears that these AMPs are minimally upregulated in the case of Mab infection of control (non Tep4 mutant) *Drosophila*. Particularly, it appears that Defensin is not upregulated by much if any by Mab infection alone (only upregulated in the absence of Tep4). The authors state that their prior (under revision) work showed that most AMPs are upregulated by Mab infection in *Drosophila*. This seems inconsistent with what is shown here (though difficult to know for sure without being able to reference the prior study). This should be more thoroughly addressed.
- Figure 4 - extent/efficacy of defensin overexpression should be shown or discussed.
- Lines 229-230 - inhibition of growth is not the same as direct bactericidal activity, results should be stated accordingly.
- Stating that defensin is "sufficient" for Mab control may be overstating the conclusions. Would rephrase to indicate that it has an important role in host defense against Mab infection.

Minor critiques

- Was a rough or smooth morphotype of Mab used? The discussion implies it was smooth but it is not otherwise discussed. Should at least be included in the materials & methods section. If smooth was used, were any studies attempted using rough morphotype?
- Lines 108-119 should be incorporated into the introduction rather than the results section
- Line 131 - it should be addressed why Relish mutant flies, but not any of the group B AMP mutants, had decreased survival.
- Figure 2B - why is Mab growth increased in the Tep5 RNAi treatment at day 5? This should be addressed.
- Figure 2D-F - should have each graph labeled with what bacteria is used.
- Figure 3I and J - should include a statistical comparison of Group A/+ and Group A/+ & Tep4>RNAi Tep4 - are these significantly different or not?

Reviewer #2 (Comments for the Author):

Touré et al., showed that AMPs were required to control extracellular Mab, and Defensin was found to be sufficient to kill extracellular Mab both in vitro and in vivo. The data suggest that Tep4-mediated opsonization of Mab allows its escape and resistance towards the Defensin bactericidal action in *Drosophila*.

Major comments:

Important comment: The authors are advised to incorporate a DOI or other relevant reference information for the unpublished manuscript "Touré et al. in revision" throughout the entirety of their manuscript. This reference contains crucial information that is essential for comprehending the current manuscript and provides evidence for the hypothesis discussed in the article. Without the DOI, I cannot presently recommend either acceptance or rejection for the manuscript.

Q1. The author demonstrated the involvement of Tep4-opsonin in the internalization of Mab into plasmacytes. However, it should be noted that *Drosophila* has six different genes for Teps, namely Tep1-Tep6, and each Tep is known to have distinct roles in promoting phagocytosis of different microorganisms. For example, TEP1 functions as an opsonin to promote phagocytosis of both Gram-positive and Gram-negative bacteria, while TEP2, TEP3, and TEP6 promote phagocytosis of *Escherichia coli*, *Staphylococcus aureus*, and *Candida albicans*, respectively, in cultured S2 cells (DOI: 10.1159/000321554). Tep5 is a pseudogene. In addition, it has been reported that Tep2 and Tep6 affect phagocytosis and melanization in flies infected with *Photobacterium* (DOI: 10.1080/21505594.2017.1330240).

In this context, it would be valuable to investigate the involvement of other Teps in Mab internalization and provide experimental results to support the findings. If there is a relevant rationale for focusing only on Tep4 among the six different Teps, the authors should describe it in the manuscript.

Q2. Figure 2A describe Tep4 involvement for internalization. However, it is not significant difference between control and Tep4 RNAi (~0.5 log₁₀ reduction) albeit Figure 2C showed significant survival difference. Thus, I feel necessity to check another Teps involeness on this Mab internalization again.

Q3. The Y-axis unit in Figure 2B should be clarified as the number of bacteria (CFU) or on a logarithmic scale if applicable.

Q4. The number of flies tested in Figure 3A is too small size. Only 5 flies look not enough for statistical analysis. The authors should use at least 20 flies for this experiment for better statistical analysis.

Q5. The authors should explain the exception of Dipteracin showing a significant difference compared to other AMPs in Figure 3E. Furthermore, also need to explain about Drosomycin and Drosocin in Figure 3F and G.

Minor comments:

Line 78: Please clarify what "Dm" stands for in the text.

Lines 79-84: The text in this section is difficult to understand and should be rewritten for clarity.

Line 126-127: Please double check the name of single gene mutation for DptSKi and AttDSKi through comparison with Hanson et al.'s study (<https://doi.org/10.7554/eLife.44341>).

Lines 125-131: It would be better provide information about gene names (single gene mutation) in an independent table to make it easier for readers to follow.

Lines 179-183: The authors should move this information to the Discussion section and include it with lines 313-318 to avoid repetition.

The statistical analysis in Figure 3I and J should be rechecked, as there appears to be a >25% difference between Tep4>RNAi Tep4 and GroupA/+ & Tep4>RNAi Tep4, but the p-value is only <0.05.

Staff Comments:

Preparing Revision Guidelines

Please return the manuscript within 60 days; if you cannot complete the modification within this time period, please contact me. If you do not wish to modify the manuscript and prefer to submit it to another journal, please notify me of your decision immediately so that the manuscript may be formally withdrawn from consideration by Microbiology Spectrum.

In this study, the authors investigate the role of antimicrobial peptides and the opsonin Tep4 in control of *M. abscessus* (Mab) infection in *Drosophila*. This manuscript builds upon the findings of a prior manuscript that is evidently in revision at present, which per the authors' description, demonstrated that Mab is internalized by *Drosophila* phagocytic plasmacytes, but was able to survive lysis of the phagocytic plasmacytes by NK cell-like thanocytes. *Drosophila* were resistant to Mab infection when thanocytes were depleted. Furthermore, they report that their previous work showed that AMPs are induced by Mab infection in *Drosophila*. In this study, they looked to build upon this finding by investigating the role of AMPs in Mab infection, and how Mab may evade the AMP response in *Drosophila*. The authors find that opsonization with Tep4 allows internalization of Mab which may allow it to evade host AMPs.

Major critiques

- Lines 133 – 134 – A major aspect of the conclusions of this manuscript seem to rest upon the concept that the Mab are extracellular when the phagocytic plasmacytes are depleted, however, this is not actually shown. Are there other cell types that could be susceptible to intracellular Mab infection in the absence of the phagocytic plasmacytes? This should be addressed as it's unclear if it already was addressed in the previous under revision manuscript or not.
- Figure 2A – the use of CFUs alone is not adequate to definitively state that Tep4 is involved in internalization. The experimental design would be unable to distinguish between internalized Mab versus Mab bound to the outer surface of the cell line. A complementary method such as flow cytometry or confocal microscopy with quenching should be utilized to make this distinction.
- Figure 2- the manuscript lacks evidence for the efficacy of the Tep4 RNAi – this should be shown in a figure or at a minimum, should be included in the text of the results section.
- Figure 2 – it is stated that Tep4 has been shown to act as an opsonin in the literature but it isn't directly demonstrated here. Results would be strengthened with addition of a western blot showing that Tep4 directly binds to Mab in the *Drosophila* model.
- Figure 3 and discussion – what is the proposed mechanism by which Tep4 depletion upregulates antimicrobial peptide expression?
- Figure 3B-D – it appears that these AMPs are minimally upregulated in the case of Mab infection of control (non Tep4 mutant) *Drosophila*. Particularly, it appears that Defensin is not upregulated by much if any by Mab infection alone (only upregulated in the absence of Tep4). The authors state that their prior (under revision) work showed that most AMPs are upregulated by Mab infection in *Drosophila*. This seems inconsistent with what is shown here (though difficult to know for sure without being able to reference the prior study). This should be more thoroughly addressed.
- Figure 4 – extent/efficacy of defensin overexpression should be shown or discussed.
- Lines 229-230 – inhibition of growth is not the same as direct bactericidal activity, results should be stated accordingly.
- Stating that defensin is “sufficient” for Mab control may be overstating the conclusions. Would rephrase to indicate that it has an important role in host defense against Mab infection.

Minor critiques

- Was a rough or smooth morphotype of Mab used? The discussion implies it was smooth but it is not otherwise discussed. Should at least be included in the materials & methods section. If smooth was used, were any studies attempted using rough morphotype?
- Lines 108-119 should be incorporated into the introduction rather than the results section
- Line 131 – it should be addressed why Relish mutant flies, but not any of the group B AMP mutants, had decreased survival.
- Figure 2B – why is Mab growth increased in the Tep5 RNAi treatment at day 5? This should be addressed.
- Figure 2D-F – should have each graph labeled with what bacteria is used.
- Figure 3I and J – should include a statistical comparison of Group A/+ and Group A/+ & Tep4>RNAi Tep4 – are these significantly different or not?

Dear Dr. Fabienne Girard-misguich:

Thank you for submitting your manuscript to Microbiology Spectrum.

Your manuscript has been evaluated by two reviewers. Although the reviewers were generally enthusiastic about the manuscript, they raised several issues that need to be resolved before we can proceed further in the editorial process. In addition, they both noted that the manuscript is largely based on another study (Touré et al. In revision) which they would like to have access to and which they think should be published before the present study. I urge you to be scrupulous in meeting this expectation.

The ASM Journals program strives for constant improvement in our submission and publication process.

Sincerely,

Olivier Neyrolles, PhD

Journals Department
Response to reviewers

Reviewer comments:

Reviewer #1 (Comments for the Author):

In this study, the authors investigate the role of antimicrobial peptides and the opsonin Tep4 in control of M. abscessus (Mab) infection in *Drosophila*. This manuscript builds upon the findings of a prior manuscript that is evidently in revision at present, which per the authors' description, demonstrated that Mab is internalized by *Drosophila* phagocytic plasmacytes, but was able to survive lysis of the phagocytic plasmacytes by NK cell-like thanocytes. *Drosophila* were resistant to Mab infection when thanocytes were depleted. Furthermore, they report that their previous work showed that AMPs are induced by Mab infection in *Drosophila*. In this study, they looked to build upon this finding by investigating the role of AMPs in Mab infection, and how Mab may evade the AMP response in *Drosophila*. The authors find that opsonization with Tep4 allows internalization of Mab which may allow it to evade host AMPs.

We would like to thank Reviewer #1 for his/her insightful comments and constructive suggestions, which we address below.

Major critiques

- Lines 133 - 134 - A major aspect of the conclusions of this manuscript seem to rest upon the concept that the Mab are extracellular when the phagocytic plasmacytes are depleted, however, this is not actually shown. Are there other cell types that could be susceptible to intracellular Mab infection in the absence of the phagocytic plasmacytes? This should be addressed as it's unclear if it already was addressed in the previous under revision manuscript or not.

Our findings in the previous paper (DOI: 10.1371/journal.ppat.1011257) indicate that phagocytic plasmacytes are the main cell population that internalizes Mab following systemic injection. This conclusion is based on the following arguments: we observed (1) an increasing number of bacteria in plasmacytes in the first 3 days of infection but not in the hemolymph, (2) a sudden bacteremia between 3 and 4 days post-infection in the recollected hemolymph, and (3) a corroboration of these observations with those made by epifluorescence microscopy. Indeed, systemic bacterial spreading was observed in *Drosophila* only after 3 days of infection.

Although phagocytic plasmacytes are the main cellular reservoir internalizing Mab, we can not exclude the possibility that other cell types could internalize the bacterium in *Drosophila*. Even if this was the case, the internalization of Mab by this potential cell population would be minor in view of the elements mentioned above.

- Figure 2A - the use of CFUs alone is not adequate to definitively state that Tep4 is involved in internalization. The experimental design would be unable to distinguish between internalized Mab versus Mab bound to the outer surface of the cell line. A complementary method such as flow cytometry or confocal microscopy with quenching should be utilized to make this distinction.

We agree with Reviewer #1 that a complementary method would reinforce the notion of Tep4 involvement in Mab internalization. However, as mentioned in the Materials and Methods section, we killed extracellular and membrane-bound bacteria with a high concentration of amikacin (250 mg/mL) for one hour before lysing the infected S2 cells. With this standardized protocol used in previous studies (Laencina *et al.*, 2018, Dubois *et al.*, 2018), we only count the intracellular bacteria.

- Figure 2- the manuscript lacks evidence for the efficacy of the Tep4 RNAi - this should be shown in a figure or at a minimum, should be included in the text of the results section.

We apologize for not including the efficacy of the two Tep4-RNAi (whole flies and S2 cells) used in the Figure 2 of the original manuscript. We have included these in the revised manuscript (pages 8 and 9). They correspond to the new Figure S1B and D, which you will find below.

- Figure 2 - it is stated that Tep4 has been shown to act as an opsonin in the literature but it isn't directly demonstrated here. Results would be strengthened with addition of a western blot showing that Tep4 directly binds to Mab in the *Drosophila* model.

We agree with Reviewer #1 that an *in vivo* demonstration of Tep4 involvement in Mab internalization would bring a plus but, unfortunately, it is technically impossible in adult *Drosophila*. Indeed, one limitation of this model is the difficulty in collecting sufficient numbers of phagocytic plasmatocytes by dissection. Tep proteins have been described as opsonins by counting percentage of internalized microbes in S2 cells using RNAi (DOI: 10.1007/978-1-59745-204-5_24). In

addition, even in larvae in which cell collection is less challenging, Tep4 involvement in *P. aeruginosa* opsonization was demonstrated by counting the fraction of bacteria internalized by plasmatocytes (DOI 10.15252/embr.201744880).

- Figure 3 and discussion - what is the proposed mechanism by which Tep4 depletion upregulates antimicrobial peptide expression?

The proposed mechanism is that Mab might be more extracellular in *Tep4*-deficient flies and thus would stimulate more Toll and Imd pathways. To test this hypothesis, we quantified the transcript levels of *Peptidoglycan recognition protein (PGRP)-SB1* and *-SD*, which encode two of the main equivalents of bacterial specific Pattern Recognition Receptors (PRR) in *Drosophila*. We found that both transcripts were upregulated. We have included these results in the revised manuscript (page 10). These correspond to the new Figure S2A and S2B which you will find below.

Figure S2

- Figure 3B-D - it appears that these AMPs are minimally upregulated in the case of Mab infection of control (non *Tep4* mutant) *Drosophila*. Particularly, it appears that Defensin is not upregulated by much if any by Mab infection alone (only upregulated in the absence of *Tep4*). The authors state that their prior (under revision) work showed that most AMPs are upregulated by Mab infection in *Drosophila*. This seems inconsistent with what is shown here (though difficult to know for sure without being able to reference the prior study). This should be more thoroughly addressed.

In our previous work (DOI: 10.1371/journal.ppat.1011257), we have observed that contrarily to *Defensin*, some AMPs-encoding transcripts (Toll-related AMPs and the Imd-related *Attacin*, *Drosocin* and *Diptericin*) are upregulated in response to Mab infection in a dose-dependent manner as observed in the Figure below. We confirmed these observations in this study, and globally, the levels of AMPs upregulation were consistent between the previous work and the present one. We have added an extra layer in this work, by showing that despite its lack of upregulation, *Defensin* is a major effector of the *Drosophila* humoral response when Mab is extracellular.

One hypothesis is that even without induction, the concentration of Defensin in the hemolymph is sufficient to kill Mab. Indeed, AMPs can have concentrations ranging from 10 to 500 μ M in the *Drosophila* hemolymph (DOI:

10.1159/000086648). Knowing that Defensin can have bactericidal action in a beginning concentration of 1 μ M (our work, DOI: 10.1159/000086648), it is possible that even without induction, its basal amount can kill Mab, which could explain the increased sensitivity of *Defensin* mutants to extracellular Mab.

- Figure 4 - extent/efficacy of defensin overexpression should be shown or discussed.

We apologize for not including the graph in the original manuscript. We have added this in the revised version which corresponds to Figure S3 that you will find below.

Figure S3

- Lines 229-230 - inhibition of growth is not the same as direct bactericidal activity, results should be stated accordingly.

The corresponding section has been rewritten in the revised manuscript (page 9) based on these suggestions.

- Stating that defensin is "sufficient" for Mab control may be overstating the conclusions. Would rephrase to indicate that it has an important role in host defense against Mab infection.

The corresponding section has been rewritten in the revised manuscript (page 9) based on these suggestions.

Minor critiques

- Was a rough or smooth morphotype of Mab used? The discussion implies it was smooth but it is not otherwise discussed. Should at least be included in the materials & methods section. If smooth was used, were any studies attempted using rough morphotype?

We apologize for not including this information in our original manuscript. In this study, we only used the smooth morphotype. We have clarified this point in the revised manuscript by naming it *S-M. abscessus*. In the future, it would be interesting to test whether the rough morphotype behaves differently; however, in this study, we would like to focus only on the smooth morphotype .

- Lines 108-119 should be incorporated into the introduction rather than the results section

We have included this in the Introduction section of the revised manuscript (page 11).

- Line 131 - it should be addressed why *Relish* mutant flies, but not any of the group B AMP mutants, had decreased survival.

As the production of Defensin is also dependent on the Imd pathway, *Relish* loss of function might impact Defensin production. The decreased survival of Imd mutant flies could be related to the latter consequence, as Defensin is important for controlling extracellular Mab. This part has been addressed in the Discussion section of the revised manuscript (page 13).

- Figure 2B - why is Mab growth increased in the *Tep5* RNAi treatment at day 5? This should be addressed.

In the new Figure 2D (former Figure 2B), Mab growth is increased at day 5 in *Tep5*-RNAi compared to *Tep4*-RNAi and not compared to the untreated control. We apologize for having present the comparison *Tep4*-RNAi versus *Tep5*-RNAi at day 5 on the original Figure. We have added the non-significant comparison between *Tep5*-RNAi and untreated cells, as for the other time points to avoid confusion for the readers.

D

- Figure 2D-F - should have each graph labeled with what bacteria is used.

The Figure has been modified accordingly.

- Figure 3I and J - should include a statistical comparison of Group A/+ and GroupA/+ & Tep4>RNAi Tep4 - are these significantly different or not?

These statistical comparisons have been added to the corresponding graphs. There is a significant difference for the Group A but not the Group C.

I

J

Reviewer #2 (Comments for the Author):

Touré et al., showed that AMPs were required to control extracellular Mab, and Defensin was found to be sufficient to kill extracellular Mab both in vitro and in vivo. The data suggest that Tep4-mediated opsonization of Mab allows its escape and resistance towards the Defensin bactericidal action in *Drosophila*.

We would like to thank Reviewer #2 for his/her insightful comments and constructive suggestions, which we address below.

Major comments:

Important comment: The authors are advised to incorporate a DOI or other relevant reference information for the unpublished manuscript "Touré et al. in revision" throughout the entirety of their manuscript. This reference contains crucial information that is essential for comprehending the current manuscript and provides evidence for the hypothesis discussed in the article. Without the DOI, I cannot presently recommend either acceptance or rejection for the manuscript.

We agree with Reviewer #2 that the paper that was in revision contains crucial information for a full understanding of the current manuscript. The paper has now been published in *PLOS Pathogens* and is accessible under DOI:10.1371/journal.ppat.1011257.

Q1. The author demonstrated the involvement of Tep4-opsonin in the internalization of Mab into plasmatocytes. However, it should be noted that *Drosophila* has six different genes for Teps, namely Tep1-Tep6, and each Tep is known to have distinct roles in promoting phagocytosis of different microorganisms. For example, TEP1 functions as an opsonin to promote phagocytosis of both Gram-positive and Gram-negative bacteria, while TEP2, TEP3, and TEP6 promote phagocytosis of *Escherichia coli*, *Staphylococcus aureus*, and *Candida albicans*, respectively, in cultured S2 cells (DOI: 10.1159/000321554). Tep5 is a pseudogene. In addition, it has been reported that Tep2 and Tep6 affect phagocytosis and melanization in flies infected with *Photographus* (DOI: 10.1080/21505594.2017.1330240).

In this context, it would be valuable to investigate the involvement of other Teps in Mab internalization and provide experimental results to support the findings. If there is a relevant rationale for focusing only on Tep4 among the six different Teps, the authors should describe it in the manuscript.

We have investigated the potential involvement of other Teps using combinations of mutations for all the four inducible Teps (from Bruno Lemaitre lab: DOI: 10.1186/s12915-017-0408-0). We observed an increased survival to Mab infection in Tep4 deficient flies, either alone or in combination with the *Tep1* homozygous mutant or with all other three inducible *Tep* homozygous mutants. The *Tep4* mutant behaved like the *Tepq^A* mutant corresponding to the combination of mutations for all inducible Teps. This suggests that the *Tepq^A* resistance phenotype is mainly conferred by *Tep4* loss of function. In combination with other data of the paper, this also suggests that Tep4 is most likely a major opsonin for Mab. These data are included in the revised manuscript (page 14) and in the new Figure 2A which you will find below.

Q2. Figure 2A describe *Tep4* involvement for internalization. However, it is not significant difference between control and *Tep4* RNAi (~0.5 log₁₀ reduction) albeit Figure 2C showed significant survival difference. Thus, I feel necessity to check another *Teps* involveness on this Mab internalization again.

The observed reduction in internalization is statistically significant, although we cannot exclude the involvement of other actors. However, the results presented in Figure 2A suggest that among the *Teps*, only *Tep4* seems to be involved in the internalization of *M. abscessus*. This decrease in 0.5 log₁₀ is in line with what is observed in the literature ((DOI:10.1016/j.chembiol.2022.03.008), (DOI:10.1016/j.isci.2023.106042) among others). This is illustrated by a study demonstrating the importance of GPL glycosylation in the internalization of *M. abscessus* where the authors observed decreases in internalization sometimes less than 0.5 log₁₀ for some mutant strains defective for GPL glycosylation. We put below a figure of the paper in question (DOI:10.1016/j.chembiol.2022.03.008).

The reduction we observed is also in the same orders of magnitude as this observed by the authors of the seminal paper describing the involvement of *Teps* in the internalization of different bacteria in S2 cells (DOI: 10.1371/journal.pbio.0040004). This is illustrated by the figure in this paper which you will find below.

Q3. The Y-axis unit in Figure 2B should be clarified as the number of bacteria (CFU) or on a logarithmic scale if applicable.

The Y-axis in the Figure 2B corresponds to a logarithmic representation of the number of bacteria per milliliter. We have renamed it “Log (CFU/mL)” to avoid confusion.

Q4. The number of flies tested in Figure 3A is too small size. Only 5 flies look not enough for statistical analysis. The authors should use at least 20 flies for this experiment for better statistical analysis.

We have counted the bacterial loads of 20 flies per condition. This result is presented in the graph below and corresponds to Figure 3A in the revised manuscript.

Q5. The authors should explain the exception of Dipterucin showing a significant difference compared to other AMPs in Figure 3E. Furthermore, also need to explain about Drosomycin and Drosocin in Figure 3F and G.

We agree with Reviewer #2 that it is surprising that, in contrast to the other tested Imd-related AMPs, *Diptericin* was less induced in infected *Tep4* mutants than in wild-type flies. This observation could be related to the isoform (*Diptericin-A*) that we amplified during qRT-PCR. We mentioned that this observation was surprising in the revised manuscript (page 10). However, since group B AMPs, to which *Diptericin* belongs, do not seem to affect *M. abscessus* even when the bacterium is extracellular, we do not feel it is necessary to investigate this difference in *Diptericin* regulation compared to other Imd-dependent AMPs.

The fact that *Drosomycin* is not upregulated on day 3 after infection with 10 CFU of *M. abscessus* is consistent with our previous work (DOI:10.1371/journal.ppat.1011257). Indeed, we only observed the induction of *Drosomycin* expression after infection with 1000 CFU.

Finally, although the difference is not huge, we observed a significant increase in the amount of *Drosocin* transcripts in the infected *Tep4* mutant compared to that in the infected control. This is consistent with what we observed with other Imd-related AMPs except for *Diptericin*.

Minor comments:

Line 78: Please clarify what "Dm" stands for in the text.

We apologize for leaving this typeface in the manuscript. This has been removed from the revised manuscript.

Lines 79-84: The text in this section is difficult to understand and should be rewritten for clarity.

The section has been rewritten for better understanding (pages 4-5).

Line 126-127: Please double check the name of single gene mutation for DptSKi and AttDSKi through comparison

with Hanson et al.'s study (<https://doi.org/10.7554/eLife.44341>).

The mutations have been correctly named in the revised manuscript (page 7).

Lines 125-131: It would be better provide information about gene names (single gene mutation) in an independent table to make it easier for readers to follow.

As requested, we have included a Table that recapitulates the different mutations in Figure 1.

A

	Group A	Group B	Group C	ΔCec^{A-C}	Rel^{E20}	spz^{rm7}	Bom^{Δ55}
	Def	AttC Dro AttA AttB Dpt AttD	Mtk Drs	CecA1 CecA2 CecB CecC	Relish	spaetzle	Bom1 Bom2 Bom3 Bom23 Bom065 Bom067 Bom068 Bom107 Bom202 Bom836
Toll	X		X	X		X	X
Imd	X	X		X	X		

Lines 179-183: The authors should move this information to the Discussion section and include it with lines 313-318 to avoid repetition.

This paragraph has been moved to the Discussion section accordingly.

The statistical analysis in Figure 3I and J should be rechecked, as there appears to be a >25% difference between Tep4>RNAi Tep4 and GroupA/+ & Tep4>RNAi Tep4, but the p-value is only <0.05.

We have checked this statistical analysis. The p-value is 0.0307. We believe that despite the difference of more than 25% on day 10, this “low” p-value may be related to the fact that the analysis takes into account the entire duration of the experiment. As observed in the figure, the difference only starts from day 6.

Staff Comments:

For complete guidelines on revision requirements, please see the journal Submission and Review Process requirements at <https://journals.asm.org/journal/Spectrum/submission-review-process>. **Submissions of a paper that does not conform to Microbiology Spectrum guidelines will delay acceptance of your manuscript.**

Please return the manuscript within 60 days; if you cannot complete the modification within this time period, please contact me. If you do not wish to modify the manuscript and prefer to submit it to another journal, please notify me of your decision immediately so that the manuscript may be formally withdrawn from consideration by Microbiology Spectrum.

May 5, 2023

Dr. Fabienne Girard-misguich
Universite de Versailles Saint-Quentin-en-Yvelines
Versailles
France

Re: Spectrum00777-23R1 (*Mycobacterium abscessus* opsonization allows an escape from the Defensin bactericidal action in *Drosophila*)

Dear Dr. Fabienne Girard-misguich:

Congratulations! Your manuscript has been accepted, and I am forwarding it to the ASM Journals Department for publication. You will be notified when your proofs are ready to be viewed.

Sincerely,

Olivier Neyrolles
Editor, Microbiology Spectrum
